# Precision Concussion Management: Approaches to Quantifying Head Injury Severity and Recovery

**DOI:** 10.3390/brainsci13091352

**Published:** 2023-09-21

**Authors:** Daniel N. de Souza, Mitchell Jarmol, Carter A. Bell, Christina Marini, Laura J. Balcer, Steven L. Galetta, Scott N. Grossman

**Affiliations:** 1Department of Neurology, New York University Grossman School of Medicine, New York, NY 10017, USA; daniel.desouza@nyulangone.org (D.N.d.S.); mitchell.jarmol@nyulangone.org (M.J.); carter.bell@nyulangone.org (C.A.B.); christina.marini@nyulangone.org (C.M.); laura.balcer@nyulangone.org (L.J.B.); steven.galetta@nyulangone.org (S.L.G.); 2Department of Ophthalmology, New York University Grossman School of Medicine, New York, NY 10017, USA; 3Department of Population Health, New York University Grossman School of Medicine, New York, NY 10017, USA

**Keywords:** concussion, traumatic brain injury (TBI), metabolic markers, neuroimaging, vision

## Abstract

Mitigating the substantial public health impact of concussion is a particularly difficult challenge. This is partly because concussion is a highly prevalent condition, and diagnosis is predominantly symptom-based. Much of contemporary concussion management relies on symptom interpretation and accurate reporting by patients. These types of reports may be influenced by a variety of factors for each individual, such as preexisting mental health conditions, headache disorders, and sleep conditions, among other factors. This can all be contributory to non-specific and potentially misleading clinical manifestations in the aftermath of a concussion. This review aimed to conduct an examination of the existing literature on emerging approaches for objectively evaluating potential concussion, as well as to highlight current gaps in understanding where further research is necessary. Objective assessments of visual and ocular motor concussion symptoms, specialized imaging techniques, and tissue-based concentrations of specific biomarkers have all shown promise for specifically characterizing diffuse brain injuries, and will be important to the future of concussion diagnosis and management. The consolidation of these approaches into a comprehensive examination progression will be the next horizon for increased precision in concussion diagnosis and treatment.

## 1. Introduction

Concussion, the mildest form of traumatic brain injury (TBI), is a widespread public health issue affecting millions annually and representing an immense economic burden to healthcare systems and nations [1]. These injuries are defined broadly as any traumatically induced transient disturbance of brain function. Given this non-specific definition, concussions can vary widely in terms of symptoms, duration, and severity [2,3]. At present, a diagnosis of concussion is largely based on subjective interpretation of physical symptoms (e.g., headache and visual disturbances), cognitive symptoms (e.g., difficulty concentrating), mood related symptoms (e.g., anxiety and depression), and behavioral symptoms (e.g., restlessness and reduced stress tolerance) [4]. Furthermore, there are at least 41 different guidelines and inventories in contemporary use for the diagnosis, grading, and classification of TBI, many of which lack empirical evidentiary support [2,5]. Many of these classification tools also tend to be broad in nature, capturing symptomology that may overlap with other conditions, such as mental health disorders, and are not applicable for specific patient cohorts, such as those with dementia [6].

A specific example of why the imprecision and subjectivity associated with concussion can be problematic is when we consider the common phenomenon of concussion in athletics; a singular, though well-known example of why a patient may be intrinsically motivated to subvert concussion detection is to remain eligible for a specific activity [7]. Several of the most common concussion symptom batteries, such as the Post-Concussion Symptom Scale (PCSS), the Standard Assessment of Concussion (SAC), and the Standard Concussion Assessment Tool (SCAT) also have inadequate reliability and validity, making them insufficient for standalone use in return-to-activity and return-to-work decisions [8]. All of this is compounded by limited treatment options and a constantly evolving understanding of concussion recovery [9,10].

The generally accepted consensus less than a decade ago was that the best course for concussion management was complete physical and cognitive rest until acute symptom resolution was achieved [9]. More recent research, however, has indicated that prolonged rest after concussion is not associated with improvement in outcomes [11]. In fact, clinical evidence has demonstrated that an early return to moderate cognitive and physical activity following a concussion is beneficial for symptom resolution and reduces the incidence of persistent post-concussion symptoms compared to prolonged rest [12,13,14,15]. This is just one example of how our understanding of concussion management is continuously evolving, and is an illustration of why increased precision in post-concussion care is critical to improving outcomes for patients with head injuries.

To address these challenges, the adoption of new and objective methods for evaluating concussion is necessary. Objective physiological markers of concussion will be essential to developing more comprehensive and standardized concussion protocols, recovery tracking, and more precise guidance about safe return to activity. This review will characterize emerging alternatives to subjective symptom questionnaires for managing suspected concussion and highlight current knowledge gaps where further research is needed.

## 2. Ocular Motor- and Vision-Based Approaches

Visual complaints are common in concussion due to damage to widely distributed vision-related neural pathways, with a reported prevalence as high as 69% in adult patients [16]. Despite the potential for objective visual measurement, standard evaluation for concussion primarily relies on self-reported somatic, cognitive, and emotional symptoms [17]. The PCSS, SCAT-5, and the vestibular ocular motor screening tool (VOMS) attempt to evaluate for visual or ocular motor symptoms to an extent; however, they still only approach this through subjective patient reporting [18,19,20]. Assessment for concussion may be improved through a broader incorporation of ocular indicators currently under investigation or in early use.

### 2.1. Convergence Insufficiency

Convergence insufficiency (CI) is a deficit in coordinated eye movement when focusing on an object at near distance [21]. The most common parameter currently measured to quantify convergence and diagnose CI is near point of convergence (NPC), which is the minimum distance from the bridge of the nose that both eyes can focus on a target without losing binocular fusion. Specific diagnostic criteria for CI are inconsistent within the existing literature, however normative NPC values tend to range from 6 to 10 cm [22]. NPC has also been shown to increase linearly with age, reaching a maximum average value of approximately 13 cm in individuals 70 and older [23]. NPC values greater than 10 cm are referred to as remote and are generally a sign of poor convergence [24].

The current literature suggests that 2% to 17% of the general population exhibits CI, compared to an estimated 49% of patients with TBI [25]. One retrospective analysis of 116 individuals, aged 5 to 21, with chronic post-concussion symptoms, observed that a receded NPC was the single most common visual impairment, present in approximately 60% of patients [26]. A systematic review of 18 studies also found moderate evidence that patients tend to have acutely impaired convergence following concussion and that deliberate ocular motor therapy after injury can improve these deficits [27].

Another recent study observed that individuals with concussion-related CI had more receded NPC values and slower convergent and divergent peak velocities compared to individuals with typically developing CI and binocularly normal controls [28]. These results suggest that concussion may influence more than NPC in terms of convergence function, and that concussion-related CI may be clinically different from other types of CI. Previous research has also shown that measurements of NPC alone may not be enough to diagnose CI, as other underlying ocular motor dysfunction may also result in clinical exam findings of receded NPC [29]. These observations indicate that more thorough and standardized diagnostic criteria are necessary for future studies to effectively capture the role of CI in concussion.

The temporal relationship between CI improvement and symptomatic recovery after concussion remains unclear. More research, including randomized controlled trials with robust methodologies and large samples, is necessary to better understand the role of convergence therapy in concussion recovery. CI also appears to manifest more significantly in younger concussion patients, so research on specific demographic features will be important in examining this measure of ocular motility dysfunction [30].

### 2.2. Accommodation

Accommodation refers to the eye’s ability to acquire and maintain focus on an object of interest with relative variation in distance via modulation of lens conformation [31]. Though there has been variability among studies regarding the reported incidence of accommodative insufficiency (AI), most estimate a prevalence of 2% to 10% in uninjured individuals under 20 years old [32,33]. Accommodative baselines tend to decrease beginning around age 30, with the most rapid declines occurring through age 50 [34]. This age-related reduction in accommodative capacity, called presbyopia, is thought to be caused by a gradual stiffening of the eye’s crystalline lens [35].

Unlike the irreversible loss of accommodation naturally associated with aging, acute onset AI has been appreciated after concussion and can result in exacerbated symptoms [30,36,37]. One study reviewed the records of 218 concussion patients and found that AI was the second most commonly reported visual deficit exhibited, with a prevalence of 42% [38]. Another study found that pediatric patients with persistent post-concussion syndrome had statistically significant reductions in accommodative amplitude (AA) compared to healthy controls and individuals with non-concussion-related CI [28]. Furthermore, numerous other studies evaluating visual disturbance after concussion have placed the incidence of AI somewhere between 42% and 74%, much higher than in the non-concussed general population [26,29,30,38].

The relationship between accommodative dysfunction and concussion is thought to be fundamentally different from that of naturally occurring age-related far-sightedness. Post-concussive AI more likely originates from deficits in control of the near triad of accommodation, convergence, and pupillary constriction, rather than physical changes in the ocular lens. In fact, abnormalities in the pupillary light reflex are understood to occur in concussion and may be associated with concussion-related photophobia [39,40,41]. Regardless of the underlying mechanism, measurement of AI has not been incorporated broadly into clinical decision making. This may be due to varied reports of normative AA, discrepant thresholds for AI diagnosis, and inconsistent methods for measuring AA [42,43,44,45,46].

Measurements of AA have potential as important objective tools in concussion care, especially in patients under 30. However, to incorporate this practically into concussion assessment, further clarity is needed on normative accommodative function [43,47]. Moreover, to make AA clinically useful, methods for measuring accommodation must be standardized.

### 2.3. Saccades and Smooth Pursuits

Saccades and smooth pursuits are examples of how the eyes acquire and maintain foveation on objects of regard. Saccades are rapid, abrupt movements which facilitate quick gaze shifts between objects [48]. By contrast, smooth pursuits allow the eyes to track objects as they move slowly and predictably through the visual field [49]. Both of these types of movements have been reported to exhibit impairments at multiple time points following concussion, though they are often subtle and challenging to detect without eye-tracking technology [50,51,52,53]. Abnormalities in saccades and smooth pursuits can be evaluated via neuro-ophthalmic examination, though significant differences exist in clinician capacity to assess for standard abnormalities like slowed saccades [54]. More recently, interest has grown in vision-based assessments using rapid automatized naming tasks like the King-Devick, Mobile Universal Lexicon Evaluation System (MULES), and Staggered Uneven Number (SUN) test [55,56,57,58]. Much attention has also been directed toward quantifying eye movements with video oculography (VOG) and applying these measurements to clinical concussion diagnosis [55,59,60,61].

When evaluated at the group level, recent studies have shown that measurements of both saccades and smooth pursuits can differentiate between subjects with mTBI and age-matched controls [62]. One study, including 28 college athletes within 72 h of diagnosed concussion and 87 healthy control athletes, found that concussion was associated with significantly increased saccadic latency and reduced saccadic accuracy when measured with VOG [63]. This study also observed that vertical smooth pursuit was more abnormal than horizontal movement in concussed individuals. This finding has been supported by at least one other study, which found that deficits in vertical smooth pursuits have some ability to discern between head injuries of varying severity, with higher prevalence in severe TBI than in moderate or mild TBI [64].

Another study, including 34 subjects with concussion and 54 controls, found that increased saccadic latency could discriminate between the groups with 92% accuracy, and that ocular measurements did not correlate with cognitive function or visual memory [65]. This finding implies that precise measurements of ocular control may yield unique clinical information, as abnormalities may be unrelated to cognitive dysfunction and specific to oculomotor pathways. Furthermore, another study observed that these types of ocular motor abnormalities strongly correlate with symptomatic burden, both acutely and chronically, post-concussion [66]. The longest case of persistent symptoms in that study was approximately 11 months post-injury. As a group, concussion subjects exhibited significant changes in smooth pursuit tracking, saccadic latency, and peak saccadic velocities, among other measures in the study [66]. Finally, recent case reports have also demonstrated that saccadic intrusions, such as opsoclonus and ocular flutter, can be associated with mTBI [67].

Because saccades and smooth pursuits are not ubiquitous or unique to concussion, the absence of these abnormalities should not independently rule out concussion [68]. Rather, observation of gross irregularities by trained clinicians should serve as an objective indicator that further concussion workup is necessary. Expertise for sideline evaluations remains a major limiting factor in scaling this type of intervention.

### 2.4. Nystagmus and Optokinetic Nystagmus

Nystagmus is an ocular motor phenomenon in which the eyes move uncontrollably in a repetitive and rhythmic manner, usually binocularly [69]. Though spontaneous nystagmus is exceedingly rare in mTBI, it can present some diagnostic value for ruling out concussion if present following a head injury. Nystagmus following head trauma can be clinically indicative of more severe injury to the brainstem, dorsal midbrain, cerebellum, or inner ear structures [70]. Based on specific characteristics of post-TBI spontaneous nystagmus, it can be diagnostically significant for localizing the injury [71,72].

One specific subtype of nystagmoid eye movement that has been investigated for its association with concussion is prolonged optokinetic after-nystagmus (OKAN) [63,66,73]. Optokinetic nystagmus (OKN) is a physiologically normal reflexive eye movement driven by rotating motion in the visual field [69]. It is characterized by smooth pursuit movements tracking a rotating focal target, such as stripes on a spinning drum, as it moves across the visual field, followed by a saccadic phase in the opposite direction toward the origin of the motion. OKAN describes the continued pattern of this reflex once the rotating visual stimulus is discontinued, and it gradually slows to a stop over seconds [73]. Initial studies have identified OKN and OKAN abnormalities in concussed populations, including reduced eye-to-stimulus velocity ratios and increased OKAN time constants [66,73]. Precise measurements of OKN and OKAN are not yet clinically practical, though initial findings support their diagnostic potential for concussion.

A brief summary of ocular motor- and vision-based approaches discussed in this section can be referenced in Table 1.

## 3. Neuroimaging in Concussion

Common diagnostic imaging techniques, such as MRI and CT, remain the first line of diagnostic imaging used by clinicians when evaluating patients with acute head injury. Most frequently, the clinical focus is on ruling out hemorrhage, fracture, or other emergent issues [74]. In less emergent outpatient settings, these types of imaging studies lack the sensitivity necessary to detect subtle abnormalities in patients with concussion [75]. Research has shown that high-resolution structural MRI can detect longitudinal changes in brain volume and cortical thickness in patients with a history of concussion; however, even high-resolution T1-weighted imaging remains largely ineffective in evaluating early concussion [76,77]. Though most structural MRI techniques are inadequate for diagnostic use in acute mTBI, several emerging techniques for identifying concussion-related changes are employed in the research domain, and have not yet become mainstays of clinical management.

### 3.1. Diffusion Tensor Imaging

Diffusion tensor imaging (DTI) is an advanced MRI technique which yields extensive information on tissue integrity and connectivity by measuring the translational diffusion of water molecules [78]. DTI maps internal structures and axonal tracts by measuring a variety of anisotropic parameters, which describe how water molecules diffuse between cells in three-dimensional space [79]. The most commonly used DTI parameters are fractional anisotropy (FA) and mean diffusivity (MD), which measure the directionality of water diffusion and the total diffusion rate, respectively, at particular points in space known as voxels. When there has been microstructural or diffuse axonal damage, FA decreases and MD increases as a result of fewer structural elements in the tissue to limit omnidirectional water diffusion [80]. In DTI, the anisotropic diffusion measurements, taken over a period of milliseconds, are synthesized into a mathematical construct called a tensor, which can be combined over an area of study to create highly detailed graphical constructs of tissues [81].

There have been a limited number of studies to specifically investigate DTI in concussion patients. One trial investigated 112 patients with recent concussion diagnoses. Investigators successfully used DTI to track significant increases in FA and decreases in MD in the corpus callosum over the course of an eight-week enhanced recovery protocol [82]. This is an important observation because it demonstrates the feasibility of DTI for monitoring post-concussion recovery. Another analysis of 53 subjects with a history of concussion and persistent cognitive symptoms identified correlations between cognitive assessment scores and DTI variable measurements (FA and MD) in frontotemporal regions of the brain [80]. Further, a third study of 11 pediatric subjects who underwent DTI within 3 days of concussion found significant increases in thalamic water diffusion anisotropy in subjects relative to controls [83]. These are important findings because they demonstrate that DTI can measure concussion severity using quantifiable surrogates for microstructural damage. They also indicate that DTI findings tend to correlate well with other metrics, such as cognitive performance and temporal proximity to injury.

One challenge with DTI is that it measures diffusion at points in space independently and cannot account for crossing axonal tracts where increased multidirectional diffusion would be expected. Because of this, areas where nerve fibers cross DTI show reduced FA values, inappropriately indicting microstructural damage. There are some more sophisticated subtypes of diffusion imaging which work based on similar principles and can resolve this issue, such as diffusion kurtosis imaging; however, these tend to come with increased expense, time, and computing demands relative to DTI [84]. Further, all current DTI research has demonstrated its ability to identify differences between groups but not individuals, as the field lacks consensus thresholds for differentiating normal and abnormal DTI parameters [85].

### 3.2. Task-Based and Resting-State Functional Magnetic Resonance Imaging

In one pooled meta-analysis of seven studies, including 174 subjects with concussion and 139 healthy controls, tb-fMRI showed objective functional differences between the two groups across a variety of cognitive tasks [86,87,88]. There was some variance in procedures and results between studies, but among them, 41 brain areas exhibited functional hyperactivation and 23 areas exhibited hypoactivation in concussion subjects. The meta-analysis revealed a single cluster of hypoactivation in the right middle frontal gyri (MFG) of the concussion group, which was present across all studies in the analysis. Using tb-fMRI, early studies have also identified functional differences in athletes with a history of concussion compared to controls, even after their cognitive performance had returned to baseline [89]. This suggests that neurological abnormalities following concussion may persist beyond subjective symptoms, and tb-fMRI may be sensitive enough to detect them.

One study, using rs-fMRI, compared 27 subjects after SRC to matched controls and demonstrated elevated global functional connectivity in the concussion group at rest [90]. Notably, the level of increased functional connectivity in the concussion subjects also correlated with recovery times and symptom persistence for athletes in the study, potentially demonstrating some level of predictive power for symptom duration. The presence of functional hyperconnectivity following concussion is consistent with extensive previous research, which has shown that both global and local increases in functional activity occur in response to neuronal insult [91]. These generalized increases in resting-state functional excitation may be a result of neurometabolic dysfunction and increased neuronal depolarization [92].

### 3.3. Dynamic Susceptibility Contrast and Arterial Spin Labeling

Dynamic susceptibility contrast (DSC) and arterial spin labeling (ASL) are forms of perfusion MRI which can be used to investigate brain activity by quantifying global and regional CBF [93]. DSC requires the injection of an exogenous contrast agent, commonly gadolinium, which travels through the brain while T2 images are collected. The gadolinium traveling with CBF influences the magnetic susceptibility of the tissues around it at a magnitude proportional to the amount of blood volume traveling to each imaging voxel [94]. ASL, on the other hand, is less invasive than DSC, as it does not require the use of exogenous contrast agents to examine CBF patterns. Instead, it uses the water content of the blood itself as the contrasting agent by inverting its magnetization as it travels through the carotid and vertebral arteries supplying the brain [95].

One study, using ASL to examine a group of 24 contact sport athletes 24–48 h post-concussion, found significant reductions in CBF compared to matched-control subjects. Specific brain areas found to have reductions in CBF compared to controls included the left inferior parietal lobule, right supramarginal gyrus, right MFG, posterior cingulate cortex, left occipital gyrus, and the thalamus, with no relative increases in CBF to any region of the brain in the SRC group [96]. Another study used DSC MRI to examine 32 subjects with persistent neuropsychiatric symptoms after a diagnosed mTBI. This study found that the mTBI group had significantly reduced global CBF compared to matched-control subjects, as well as specific regional CBF abnormalities that were statistically associated with more severe psychiatric symptoms [97].

One pooled meta-analysis of 23 studies using ASL to evaluate 566 concussion patients found reasonably strong evidence that mTBI is associated with CBF abnormalities, which somewhat correlate with clinical recovery [98]. The preponderance of current evidence appears to indicate a general trend of decreased global CBF in the acute, subacute, and chronic stages of recovery. However, there is significant heterogeneity among previous studies in terms of methodology and reported results. Most existing studies also have sample sizes of fewer than 50 subjects and look at widely varying mechanisms of injury and time-points of recovery. With that said, we can be fairly confident that CBF abnormalities do occur following diffuse head injuries, but the clinical applicability of this finding remains unclear.

Though there is much more work to be conducted in order to characterize changes in CBF after concussion, we do have some guidelines regarding what is abnormal, which can be applied to individual patients and future studies. In healthy adults, global CBF is approximately 50 mL of blood per 100 g of cerebral tissue per minute (mL/100 g/min), with values below 20 mL/100 g/min typically leading to neural dysfunction [99,100,101]. Future studies should aim to determine which characteristics of head injury correlate with acute CBF values that meet this threshold and how they influence long term recovery patterns.

### 3.4. Susceptibility-Weighted Imaging

When the brain experiences trauma, the cortex often moves with different speed and momentum than the subcortical white matter, which induces shearing forces that can lead to axonal and blood vessel stretching and damage. The diffuse injury associated with acceleration-related trauma often results in microscopic bleeding in the brain’s vasculature, much of which lies below the detection threshold of standard imaging [102]. Susceptibility-weighted imaging (SWI) is a form of MRI that is sensitive to the differences in magnetic properties between tissues and uses that information to generate uniquely contrasted anatomical images [103]. Because of the strong magnetic properties of iron, SWI can be effective in the acute and long-term periods for detecting cerebral micro-hemorrhages caused by this shearing stress [104].

SWI has been observed to detect up to six times as many hemorrhagic lesions and to have increased sensitivity to smaller-sized (<10 mm) lesions in the brains of moderate-to-severe TBI patients compared to more traditional T2-weighted MR images [105,106]. It has also demonstrated particular efficacy in detecting brainstem microhemorrhages compared to other imaging types [107]. Specifically, in mTBI and concussion, microvascular hemorrhagic injuries have been detected with SWI in 22% to 28% of mTBI subjects, compared to 0% to 10% in non-head-injury control subjects, in both the acute and subacute periods [108,109,110,111].

Several studies have also established an association between the total post-concussion microhemorrhage burden (total number of hemorrhages and cumulative volume) and acute post-injury Glasgow Coma Scale (GCS) scores, as well as neurological disability and cognitive impairment in both the short and long term following injury [110,112,113,114,115]. This may indicate that SWI has potential viability for stratifying concussion severity and could serve as an independent predictor of recovery outcomes. Notably, some studies have shown cerebral microhemorrhages to be some of the most persistent manifestations of concussion and TBI, lasting up to five years post-injury, with apparent associations with persistent symptoms [116,117].

Research is still necessary to determine the long-term impacts of cerebral microbleeds that persist for months to years. Some research has associated microbleeds with adjacent areas of cell loss and demyelination, raising questions about the risk profile of long-term disability, cognitive decline, and associations with other cerebral pathophysiology [111].

### 3.5. Magnetic Resonance Spectroscopy

Magnetic resonance spectroscopy (MRS) is a relatively simple and non-invasive way to measure the metabolite content of tissues [118]. Proton (^1^H) MRS, the most common modality, can be conducted on all commercial magnetic resonance imaging (MRI) scanners at standard field strength (3T), or higher, depending on the protocol. Information garnered can be integrated into T_1_- and T_2_-weighted MR images or interpreted independently via graphical spectra [119]. Major metabolic products of interest that can be detected in tissues with MRS include NAA, choline, creatine, lactate, lipids, glutamate, glutamine, gamma-aminobutyric acid (GABA), myo-inositol, and certain amino acids, depending on the strength of the magnetic field and acquisition protocol [120,121].

The sensitivity of ^1^H MRS for detecting biochemical abnormalities associated with TBI has been consistently observed in numerous studies dating back to 1998 [122,123,124,125]. The most consistent observation of these studies has been reduced amounts of NAA in neural tissues, measured as both absolute concentration and ratio to other brain metabolites [121]. In a 2022 meta-analysis of 138 studies, Joyce and colleagues observed that, among 1428 brain-injured subjects, the corpus callosum was the region most susceptible to metabolic changes, and that NAA was consistently lowered in patients with TBI of any severity [126]. This may be due to structural vulnerability to shear injury. In addition to the corpus callosum, NAA values were observed to be significantly lower in the frontal, parietooccipital, and parietal regions, and no region of the brain exhibited any increase in NAA concentrations following injury.

NAA is one of the most highly concentrated molecules in CNS and has the most prominent proton signal in ^1^H MRS, making it one of the most reliable markers that can be observed in MRS studies [127]. It is a common representative measure of neuronal health because it is found almost exclusively within neurons. Fluctuations in NAA concentration are thought to reflect changes in neuronal structural integrity. Further, NAA is synthesized in the neuronal mitochondria and may have a role in energy metabolism, so a reduced concentration may be an indirect representation of the energy impairments thought to occur following brain injury [92,127,128]. NAA concentration changes have also exhibited normalization with clinical recovery in mTBI, meaning that they may have validity as objective indicators of recovery for return-to-play decisions after SRC [127,129].

Despite relatively consistent evidence that ^1^H MRS can detect biochemical differences between concussed and control subjects, integration into clinical practice remains limited. Like several other concepts discussed in this review, no precise definition of “abnormal” is currently accepted, and causality between MRS aberration and concussion remains speculative. Future research should aim to further characterize acute post-injury and recovery profiles of NAA and other metabolites in MRS.

### 3.6. Emerging and Unexplored Concepts

Ultra-high-field MRI typically uses 7 Tesla magnetic field strength with existing MRI modalities to display highly detailed images with significantly improved signal-to-noise ratios, capable of elucidating subtle anatomical abnormalities [130]. Currently, a very limited number of studies have employed this tool in the examination of TBI and concussion [131,132,133], largely due to the significant cost and limited availability of such techniques. However, since the U.S. Food and Drug Administration approved the first 7T MRI machine in the U.S. in 2017, this capability has continued to grow [134]. High-field MRI capability may prove to be significant in discovering subtle white and grey matter changes in the brains of concussion patients, and future studies should aim to employ them in investigations. 

Artificial neural networks have shown significant promise in the area of interpreting medical imaging studies and have been a topic of immense interest in the recent medical literature [135]. Large language, image, and data processing models have the ability to synthesize and apply immense amounts of information in incredibly short periods of time, making them increasingly more essential tools in modern data analysis [136]. Based on this review, one of the largest problems with our current understanding of imaging patterns after concussion is that they have largely only demonstrated efficacy for comparisons at the group level, and the thresholds necessary to apply them to individual patients do not exist yet. One specific application for artificial intelligence, which could rapidly change how concussion and TBI are managed, is to use it to identify specific imaging parameters associated with concussion outcomes [136].

A summary of the imaging methods discussed in this section and their demonstrated applications for concussion can be referenced in Table 2.

## 4. Biochemical Concentrations in Blood and Other Tissues

Following head injury, myriad systemic biochemical responses occur that are not limited to the central nervous system (CNS) [92,137]. Several blood- and other tissue-based biomarkers have been studied for predictable associations with concussion, including glial fibrillary acidic protein (GFAP), ubiquitin C-terminal hydrolase-L1 (UCH-L1), the protein S100β, tau, neurofilament light (NFL), N-acetyl aspartate (NAA), and systemic inflammatory markers [121,138,139,140,141,142]. These specific biomarkers have been observed to accurately distinguish individuals with intracranial injury from healthy controls with a high degree of sensitivity and specificity [143,144,145].

GFAP is an intermediate filament protein expressed in the astrocytes of the CNS, where it serves as the primary cytoskeletal structure for these glial cells [146]. Levels of this protein have been observed to elevate in the blood within one hour of head trauma and can reach peak levels between 6 and 24 h after injury [147]. They also appear to maintain statistically significant elevations for anywhere from 3 to 30 days following injury [148,149]. Some studies have suggested that acute concentrations of serum GFAP correlate with injury severity, in addition to cognitive performance and clinical outcomes at multiple time-points following concussion [150,151,152,153]. GFAP may be more relevant than other biomarkers for cases in which patients delay seeking care due to prolonged clearance from serum [148].

UCH-L1 has also been studied for its role after head injury. Several studies have demonstrated that blood levels of UCH-L1 peak very early after head trauma and decrease rapidly, unlike GFAP, which tends to gradually rise over the first 24 h post-injury [148,153]. Day-of-injury measurements of UCH-L1 have demonstrated substantial prognostic value for predicting injury severity, the presence of CT lesions, and the quality of clinical outcomes [150,153,154,155]. In fact, the U.S. Food and Drug Administration has already approved, for clinical use, a blood test which measures circulating UCH-L1 and GFAP levels, to help determine the need for computed tomography (CT) scans within 12 h of head injury [156]. However, challenges remain in terms of widespread availability and application of these assays.

Other compounds which have been shown to demonstrate identifiable abnormalities in blood concentrations following brain injury include the calcium-binding protein S100β, the axonal protein NFL, and the structural protein tau [138,140,142]. Adding to distinct patterns of elevation in acutely concussed patients, each of these has also exhibited some level of association with mortality, symptom duration, and functional outcomes at multiple points during recovery, giving them meaningful prognostic value [142,157,158,159].

In addition to these specific biomarkers, inflammation is also known to play a critical role in the pathophysiology of concussion [92]. For example, increases in concentrations of inflammatory markers, such as interleukin-6 (IL-6), interleukin-1 receptor antagonist (IL-1 RA), and C-reactive protein (CRP), among others, have been associated with the acute response to concussion, as well as being somewhat predictive of symptom duration and recovery [160,161,162]. These inflammatory markers can be non-specific, so the extent of their value in concussion management is currently unclear.

A summary of significant blood- and other tissue-based metabolites associated with concussion discussed in this section can be referenced in Table 3.

## 5. Limitations

This review has several associated limitations. First and foremost, our review is not meant to serve as an absolutely comprehensive catalog of the existing literature on objective concussion evaluation. Rather, this work examines a representative snapshot of some of the most promising areas of concussion research and highlights specific shortcomings and directions for further investigation. The topics discussed in this review were selected based on documented evidence from a number of recent peer-reviewed studies demonstrating potential efficacy for use in concussion care. Furthermore, it is acknowledged that this is a rapidly evolving field, and that some newer and developing topics with smaller bodies of supporting evidence have been excluded in favor of a discrete number of topics with more robust evidentiary support.

Another limitation of this review is one that is inherent to the nature and evidentiary state of the topics discussed. Specifically, all of the discussed methods have shown some level of efficacy for discerning between groups of concussed and non-concussed individuals, but few have specific clinical thresholds to define what is abnormal on an individual level. Due to this fact, this review has refrained from analyzing the accuracy, specificity, and sensitivity of these individual diagnostic modalities in any great depth. Fully understanding the accuracy, specificity, and sensitivity of many of these methods is beyond what can reasonably be accomplished at this point in time without more in-depth investigations and data. However, what the authors of this review can assess from the current literature is that none of these methods will likely ever be completely sufficient for concussion evaluation on their own.

## 6. Directions for Further Investigation

The ultimate goal for applying the diagnostic modalities discussed in this paper is to incorporate them into a formulaic, comprehensive progression of care for individuals with traumatic head injuries. This type of model will rely on combining inexpensive, easily administered diagnostic tests, such as the discussed ocular motor tests, with more advanced and resource-intensive methods, such as blood concentrations and imaging modalities, in a progressive manner based on injury severity and symptom persistence. A comprehensive model that accounts for all of these different approaches, in combination with the current standard of symptom-based evaluations, would allow for an easily modulated, yet standardized, care model that is based on objective, quantifiable data. Over time, this model will become modifiable for use with all concussion patients, regardless of severity, demographics, or concussion etiology, as data are continually gathered and synthesized.

However, before this can become a reality, more research is necessary in order to identify specific thresholds, protocols, and guidelines that can be applied to individuals in the clinical setting. The goal of future research studies in this area should be to specifically define thresholds for concussion in different patient cohorts, using the modalities discussed in this review. Future research efforts should prioritize conducting this in diverse cohorts of patients, especially cohorts that are varied in age and concussion etiology, in order to make a broadly applicable model that is possible to develop and apply clinically.

## 7. Conclusions

Mitigating the detrimental effects of concussion and TBI is a particularly difficult public health challenge. Firstly, it is prevalent, and because the mechanisms involved are sporadic and traumatic in nature, prevention can be challenging. Secondly, the clinical definition of concussion is broad and non-specific, which results in highly variable presentation and imprecise approaches to clinical management. To compound these factors, much of contemporary concussion management relies on subjective symptom reports, which may be influenced by any number of confounding elements for each patient. These issues will likely always exist, due to individual biological inconsistencies across patients and variable mechanisms of injury. However, preliminary research on objective physiological markers of concussion indicates that there is room for improvement in how concussion is identified, stratified, and managed. Objective clinical assessments of visual and ocular motor symptoms, specialized imaging techniques, as well as blood and other tissue-based metabolite concentrations have all shown promise for characterizing brain injuries and will be important to the future of concussion care. Consolidating a selection of these tools, through further study and the use of artificial intelligence, into a comprehensive clinical exam of objective measures is necessary to make concussion care more precise and tailored to each unique patient.

## Figures and Tables

**Table 1 brainsci-13-01352-t001:** Summary of ocular motor- and vision-based approaches to evaluating concussion.

Visual/Ocular Motor Measure	Distinctive Factors for Concussion	Advantages	Limitations
Convergence Insufficiency	▪ 2–17% general population prevalence ▪ 49–60% incidence after concussion	▪ Easily and quickly measured ▪ Applicable to individual patients	▪ Less applicable to older populations
Accommodative Insufficiency	▪ 2–10% general population prevalence ▪ 42–74% incidence after concussion	▪ Strong potential for application in young patients with no history of far-sightedness	▪ Less applicable for older populations ▪ Less applicable with history of far-sightedness ▪ Varied reports of normative values ▪ Varied and technical means of measurement
Saccades and Smooth Pursuits	▪ Slowed saccades ▪ Increased saccadic latency ▪ Reduced saccadic accuracy ▪ Abnormal vertical smooth pursuits ▪ Slowed pursuits	▪ Significant abnormalities detectable at bedside via neuro-ophthalmic exam ▪ Proxy measurements possible through easily administered tests ▪ (KD, MULES, SUN) ▪ May provide unique clinical information independent of cognitive function	▪ Requires trained clinician exam ▪ Subtler abnormalities require eye recording equipment to detect
Optokinetic Nystagmus	▪ Reduced eye-to-stimulus velocity ratio ▪ Lengthened OKAN	▪ Reflexive and completely objective	▪ Difficult to measure ▪ Not yet clinically applicable

**Table 2 brainsci-13-01352-t002:** Summary of imaging modalities with demonstrated applicability for concussion.

Imaging Method	Measured Quantity	Applications for Concussion Demonstrated by Previous Research
Diffusion Tensor Imaging	Microstructural and diffuse axonal damage via directionality and rate of cerebral water diffusion	▪ Improvement tracking to monitor progress with standardized recovery protocol ▪ Accurately differentiates between acutely concussed patients and controls based on increased cerebral water diffusion ▪ Accurately differentiates between chronic post-concussion syndrome subjects and controls based on increased cerebral water diffusion
Task-Based and Resting-State Functional MRI	Localized cerebral hyperactivation, hypoactivation, and functional connectivity via oxygen concentration gradients	▪ Provides clinically unique information independent of cognitive performance in subjects with history of concussion ▪ Identifies consistent patterns of localized abnormalities associated with concussion ▪ Identifiable patterns of increased functional connectivity shown to have predictive power for symptom duration
Dynamic Susceptibility Contrast and Arterial Spin Labeling	Localized and global changes in cerebral metabolic activity via cerebral blood flow and changes in magnetic properties	▪ Accurately differentiates between acutely concussed patients and controls based on reduced cerebral blood flow measurements ▪ Accurately differentiate between chronic post-concussion syndrome subjects and controls based on reduced cerebral blood flow measurements
Susceptibility-Weighted Imaging	Cerebral microhemorrhages via magnetic susceptibility differences between tissues	▪ 0–10% prevalence of cerebral microhemorrhages in uninjured population compared to 22–28% incidence after concussion ▪ Total micro-hemorrhage burden shown to correlate with outcomes after head injury
Magnetic Resonance Spectroscopy	Biochemical composition of different tissues based on magnetic properties	▪ Identifies consistent patterns of localized abnormalities in head injury ▪ Has exhibited normalization with clinical recovery, presenting viability as an objective indicator of recovery ▪ Accurately differentiates between acute concussion and controls ▪ Accurately differentiates between chronic post-concussion syndrome subjects and controls ▪ Measures metabolic products which may serve as indirect measures of neuronal structural damage and cerebral energy impairments

**Table 3 brainsci-13-01352-t003:** Summary of blood- and other tissue-based metabolites indicative of concussion.

Significant Metabolite	Change Associated with Concussion	Detection Modality	Advantages Demonstrated by Previous Research
Glial Fibrillary Acidic Protein	↑	Serum concentration	▪ Predictive of CT lesions if taken acutely ▪ Demonstrates prolonged serum clearance time and may have application for patients who delay seeking care ▪ Absolute acute concentrations correlate with clinical outcomes
Ubiquitin C-Terminal Hydrolase-L1	↑	Serum concentration	▪ Predictive of CT lesions if taken acutely ▪ Absolute acute concentrations correlate with clinical outcomes
N-Acetyl Aspartate	↓	Magnetic Resonance Spectroscopy	▪ Accurately differentiate between control subjects and subjects with TBI of any severity ▪ Exhibit normalization with clinical recovery
S100β	↑	Serum concentration	▪ Acute concentrations correlate with mortality, symptom duration, and functional outcomes ▪ Differentiate between acutely concussed and control subjects
Neuro- filament Light	↑	Serum concentration	▪ Acute concentrations correlate with mortality, symptom duration, and functional outcomes ▪ Differentiate between acutely concussed and control subjects
Tau	↑	Serum concentration	▪ Acute concentrations correlate with mortality, symptom duration, and functional outcomes ▪ Differentiate between acutely concussed and control subjects

↑, ↓: The serum concentration of this metabolite either increases or decreases with mTBI.

## Data Availability

No new research data were created or analyzed during the conduct of this review study.

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
