# Peer review of "Precision Concussion Management: Approaches to Quantifying Head Injury Severity and Recovery"

_brainsci, 2023, doi:10.3390/brainsci13091352_

Round 1
Reviewer 1 Report
This is a review of select topics in concussion. The stated purpose is to summarize objective tests used for concussion. I have a few comments about this manuscript:
- It might be useful to better clarify the concussion patients served by these tests. Is the focus athletes? Or all comers? Should there be a difference?
- The paper would be more readable if it was more focused. There is a lot of general information about concussion that detracts from the article's stated purpose. The sections of the manuscript are diverse and the subsections do not interrelate well.
- It would be helpful if there was more interpretation of the reviewed studies with respect to what should be done in the future. What is a promising avenue, what is not, what place might each test have in diagnosis and/or management?
- Ocular Motor and Vision-Based Approaches. The authors should better delineate the problems with VOMS and how these other approaches may be better. It is based primarily on symptoms, but it is a test rather than a survey. Table 1 is basically VOMS.
- The MRI section is a summary of studies on MRI in concussion. There is not much interpretation of the findings. MRI in concussion is entirely academic right now, and it is difficult to imagine how to scale these efforts. Do the authors suggest MRI be used routinely in concussion?
- Biochemical tissue concentrations seems an odd way to describe blood tests. The only non-blood test is MRS which was discussed in the imaging section.
It is fairly well written, a few errors here and there.
Author Response
Dear reviewers,
Thank you so much for the time and effort that you put into reviewing our work and for the thoughtful comments you have provided. Responses to the points you provided can be referenced below:
Reviewer 1:
- It might be useful to better clarify the concussion patients served by these tests. Is the focus athletes? Or all comers? Should there be a difference?
The topics discussed in our paper are not meant to be specific to any individual group of patients. However, at specific points in the paper we discussed sport-related concussion as an example where a topic may demonstrate particular applicability or where diagnostic imprecision is especially problematic. In response to your comment we have modified a statement (line 41-44) and added a sentence (line 494-495) to highlight that.
- The paper would be more readable if it was more focused. There is a lot of general information about concussion that detracts from the article's stated purpose. The sections of the manuscript are diverse and the subsections do not interrelate well.
Thank you for the critique. The nature of current research on objective concussion markers is such that the diversity of our paper subsections is necessary. Based on our review of the literature, the discussed topics appear to be the most advanced in terms of their potential applicability in concussion diagnostics, and therefore all warrant discussion.
- It would be helpful if there was more interpretation of the reviewed studies with respect to what should be done in the future. What is a promising avenue, what is not, what place might each test have in diagnosis and/or management?
In response to your recommendation, the authors have added two sections to the paper discussing limitations of the current discussion and recommended directions for further research (lines 459-502).
- Ocular Motor and Vision-Based Approaches. The authors should better delineate the problems with VOMS and how these other approaches may be better. It is based primarily on symptoms, but it is a test rather than a survey. Table 1 is basically VOMS.
Thank you for your recommendation. The authors specifically mention VOMS on line 71 of this review and briefly state that the problem with the test lies in its reliance on subjective patient report, relative deviations from individual baselines which are often absent, and test-induced symptom aggravation. The main weakness of VOMS lies in that it lacks both internal and external controls, resulting in limited reliability over time. This is in contrast to things like RAN tasks and numerical measurements which have a standard outcome and are more reliable across administrations. What the authors are proposing is to utilize specific measurements for each of the tests in VOMS and compare them to aggregate data in order to assess for objective, as opposed to relative, abnormalities.
- The MRI section is a summary of studies on MRI in concussion. There is not much interpretation of the findings. MRI in concussion is entirely academic right now, and it is difficult to imagine how to scale these efforts. Do the authors suggest MRI be used routinely in concussion?
Thank you for your question. The authors of this paper are not suggesting that MRI be implemented routinely in concussion work up at this time with the current state of our understanding. In this paper, we presented existing evidence that suggests the discussed strategies may be useful for quantifying and stratifying concussion severity as well as tracking recovery. The key shortcoming we highlighted in the paper is that, while promising, specific thresholds, protocols, and guidelines which could be applied to individuals in the clinic do not yet exist. Please see the addition of our added limitations section which states that more research is necessary to define these thresholds and incorporate them into a comprehensive clinical progression before they can be applied clinically.
- Biochemical tissue concentrations seems an odd way to describe blood tests. The only non-blood test is MRS which was discussed in the imaging section.
Thank you for the observation. We used the term “tissue concentrations” with specific consideration for our discussion of MRS. We did discuss MRS in the imaging section, as we felt it was more appropriately placed there, however we also chose to include it in table 3, as it was also relevant to that discussion. In response to your comment we have changed the title of section 4 to place more of an emphasis on blood concentrations.
Reviewer 2 Report
This is a well-written and well-argued review of the need for precision concussion measures. The authors examined candidate measures in three domains: 1) oculomotor biomarkers, 2) radiological biomarkers, and 3) blood biomarkers. These biomarkers could serve three different purposes: 1) Differentiation of concussed individuals from non-concussed individuals, 2) quantifying the severity of the concussion, and 3) serving as prognosticators for the outcome. As discussed by the authors, no one biomarker does all three things with accuracy and precision. There remains a continuing need to identify biomarkers to make precision concussion management a reality. The authors use many abbreviations, and a table of abbreviations would enhance the paper. I would suggest adding a few sentences at the end of the discussion as to the limitations of the paper (i.e. list of references is selective and not comprehensive (i.e. representative). Also authors made no attempt to quantify the accuracy, specificity, or sensitivity of any of the proposed metrics ( a necessary element for any metric selected for precision concussion, e.g. AUC, etc.).
Author Response
Dear reviewers,
Thank you so much for the time and effort that you put into reviewing our work and for the thoughtful comments you have provided. Responses to the points you provided can be referenced below:
Reviewer 2:
This is a well-written and well-argued review of the need for precision concussion measures. The authors examined candidate measures in three domains: 1) oculomotor biomarkers, 2) radiological biomarkers, and 3) blood biomarkers. These biomarkers could serve three different purposes: 1) Differentiation of concussed individuals from non-concussed individuals, 2) quantifying the severity of the concussion, and 3) serving as prognosticators for the outcome. As discussed by the authors, no one biomarker does all three things with accuracy and precision. There remains a continuing need to identify biomarkers to make precision concussion management a reality. The authors use many abbreviations, and a table of abbreviations would enhance the paper.
Thank you for your comments. We have added a table of all abbreviations used in the paper to the end of the paper, before the reference list.
I would suggest adding a few sentences at the end of the discussion as to the limitations of the paper (i.e. list of references is selective and not comprehensive (i.e. representative).
Thank you for this suggestion. We have added a new section to the paper in response to your recommendation. Please reference lines 459-483.
Also authors made no attempt to quantify the accuracy, specificity, or sensitivity of any of the proposed metrics ( a necessary element for any metric selected for precision concussion, e.g. AUC, etc.).
Thank you for this suggestion. We have addressed this shortcoming of our review specifically in lines 474-478. The state of the current research on these topics is such that the majority of them have only demonstrated use thus far in differentiating between groups, therefore it would be premature to specifically make claims regarding accuracy and specificity of all discussed topics.
Reviewer 3 Report
Concussion is one of the manifestations of mild traumatic brain injury, and its diagnosis lacks objective criteria, and there have been many studies looking for objective diagnosis methods over the years. This review comprehensively expounds and summarizes the current researches, which is commendable. Meanwhile, I have some suggestions for authors as follows:
1. This review focuses on visual and ocular motor concussion symptoms and specialized imaging techniques, but the summary of biological tissue markers is not comprehensive enough. In recent years, the number of studies on concussion molecular biomarkers has increased significantly, more than imaging markers methods.
2. It is suggested that the author try to propose feasible diagnostic methods for concussion based on the summary of existing studies.
3. The author may also try to summarize the shortcomings of the current researches and the directions worthy of further exploration.
Author Response
Dear reviewers,
Thank you so much for the time and effort that you put into reviewing our work and for the thoughtful comments you have provided. Responses to the points you provided can be referenced below:
Reviewer 3:
Concussion is one of the manifestations of mild traumatic brain injury, and its diagnosis lacks objective criteria, and there have been many studies looking for objective diagnosis methods over the years. This review comprehensively expounds and summarizes the current researches, which is commendable. Meanwhile, I have some suggestions for authors as follows:
- This review focuses on visual and ocular motor concussion symptoms and specialized imaging techniques, but the summary of biological tissue markers is not comprehensive enough. In recent years, the number of studies on concussion molecular biomarkers has increased significantly, more than imaging markers methods.
The authors agree that the discussions in all three of the main topical sections of this paper are discrete in nature and not completely comprehensive. For the blood-based biomarkers specifically, the intent was to discuss a select number of specific markers with substantial evidentiary support and which will be feasible for clinical application in the shortest timeline. We understand your concern with this approach, and have addressed this limitation in lines 460-469.
- It is suggested that the author try to propose feasible diagnostic methods for concussion based on the summary of existing studies.
Thank you for this suggestion. In response to your comment we have discussed, in theoretical terms only, an idea of what a future model might look like in lines 488-491. All of the diagnostic techniques discussed in this paper have yet to demonstrate diagnostic feasibility on an individual basis, and proposing a specific concussion diagnostic model is outside the scope of this review.
- The author may also try to summarize the shortcomings of the current researches and the directions worthy of further exploration.
Thank you for this suggestion. In response to your comment we have added two new sections to the paper. Please reference lines 459-503.
Reviewer 4 Report
As a prevalent condition, the diagnosis of concussion is mostly based on presenting symptoms, which also dictates the management course of it. Multiple factors can influence our understanding of the patient state and the course of action required. Recently, advances have helped us to have a better understanding of the mechanism of concussion, and therefore, diagnosis and treatment that are required. In this review, the authors investigated the emerging approaches to evaluating concussion. This is an interesting review which can add to the body of our knowledge on an important topic, but there are comments which need to be addressed before it is considered further. I provided detailed major and minor comments below:
1. Abstract: Line 13: “These types of reports may be influenced by a variety of factors for each individual …” Please provide some examples.
2. The abstract should be more specific. Discussed approaches, such as ocular movement, neuroimaging, biochemical, etc. should be mentioned.
3. Abstract: Line 15: This would have been comprehensive/thorough if the manuscript was a systematic review/meta-analysis of the literature. To avoid confusion, please remove the word “thorough”
4. Introduction: “Many of these classification tools also tend to be broad in nature, capturing symptomology that may overlap with other conditions, such as mental health disorders” Please mention that there are no guidelines for specific patient cohorts either, such as those with dementia and mental health disorders, etc, therefore TBI in these groups can be more challenging to diagnose and treat.
5. Introduction: Please mention the wider implications of TBI, in terms of healthcare costs, lost labor, etc
6. Introduction: Line 33: “Many of these classification tools also tend to be broad in nature” Is this an issue with the classification or the nature of concussion which is not associated with a specific and unique symptomatic presentation? Of course having a precise sign/symptom would be ideal, but how possible is identifying and implementing such specific signs/symptoms in the healthcare setting? I suggest caveats from a clinical perspective should not be mixed with logistics caveats.
7. Introduction: Lines 36-28: “The diagnostic imprecision and subjectivity associated with concussion can be especially challenging when considering concussion in athletics, as many athletes may be motivated to subvert concussion detection to remain eligible for participation[7].” This sentence is out of place and it is not clear why athletes are highlighted and focused. It may mislead the reader that the manuscript is about concussions in athletes. If the authors would like to mention athletes, they need to mention other cohorts as well to provide a balanced argument.
8. Introduction: Lines 41-42: “… also have imperfect reliability and validity, making them inadequate for standalone use in return-to-play decisions” Please provide more specific examples to make such deficiencies clear.
9. Line 41: Please replace “imperfect” with “inadequate” as perfection is rarely achieved in clinical settings, and in the semantic discussion, perfection can be quite subjective!
10. Introduction: Please mention briefly the signs and symptoms of concussion.
11. Line 51: “post-concussion symptoms” Please provide more specific examples
12. Lines 63-64: “Visual complaints are common in concussion, with a reported prevalence as high as 69% in adult patients” What is the mechanism of injury causing visual complaint in concussion? To achieve a thorough understanding, explaining such mechanisms can help establish diagnostic and prognostic guidelines.
13. Line 66: remove “all”
14. Line 76: “. Specific diagnostic criteria for CI are inconsistent within the existing literature” If such inconsistencies exist, how is this going to be incorporated into concussion diagnosis? Please clarify
15. Line 180: Nystagmus can also overlap with other conditions. Rightly, the authors criticized that some current criteria for concussion can overlap with other conditions, such as dementia. Similarly, nystagmus can overlap with other conditions. So how is the alternative better? Please clarify.
16. Table 1: Adding a column titled “References” to mention papers that provided information can help readers trace the information.
17. Lines 210 and 212: using two “however” does not help the flow. Please rewrite this section.
18. Line 243: “One challenge with DTI …” Another challenge of DTI is accessibility to such technology in low and middle income countries, where TBI and concussion is more prevalent. So to achieve a universal patient outcome, such factors need to be considered.
19. Adding a figure to summarize the neuroimaging in concussion approaches, such as DTI, tb-fMRI, DSC, etc, can improve the quality of the manuscript. Table 2 is fine, but the manuscript needs some figures to visualize the discussed concepts.
20. Line 392: The usage of AI is revolutionizing the healthcare system and mentioning such an important topic is crucial for this manuscript, so I command the authors for mentioning this topic.
21. Line 392-393 “Artificial neural networks have shown significant promise in the area of interpreting medical imaging studies and have been a topic of immense interest in the recent medical literature.” This sentence needs a citation and the following paper can be used:
Neurosurgery and artificial intelligence. AIMS Neurosci. 2021 Aug 6;8(4):477-495. doi: 10.3934/Neuroscience.2021025. PMID: 34877400
22. Biochemical Tissue Concentrations section is significantly shorter compared to the other two sections, whereas there is plenty of evidence to expand on.
23. Table 3: Abbreviations should be added to the Table footnote.
24. Table 3: Adding two columns mentioning the advantages and disadvantages of each metabolite can be helpful.
25. Similarly, information in Figure 3 can be summarized into a figure.
26. Line 430: “FDA” has been used in line 388, whereas the abbreviation is introduced in line 430. Care should be taken.
27. Conclusion: Line 453: I would suggest removing the word “extremely” as it is not the case.
28. Line 465: “Consolidating a selection of these tools…” The usage of AI can be highlighted here for making diagnostic and prognostic models.
Some minor edits are required as mentioned.
Author Response
Dear reviewers,
Thank you so much for the time and effort that you put into reviewing our work and for the thoughtful comments you have provided. Responses to the points you provided can be referenced below:
Reviewer 4:
- Abstract: Line 13: “These types of reports may be influenced by a variety of factors for each individual …” Please provide some examples.
Reference line14-16 in response to your comment.
- The abstract should be more specific. Discussed approaches, such as ocular movement, neuroimaging, biochemical, etc. should be mentioned.
Reference line 19-21 in response to your comment.
- Abstract: Line 15: This would have been comprehensive/thorough if the manuscript was a systematic review/meta-analysis of the literature. To avoid confusion, please remove the word “thorough”
Thank you for this recommendation. The word “thorough” has been removed.
- Introduction: “Many of these classification tools also tend to be broad in nature, capturing symptomology that may overlap with other conditions, such as mental health disorders” Please mention that there are no guidelines for specific patient cohorts either, such as those with dementia and mental health disorders, etc, therefore TBI in these groups can be more challenging to diagnose and treat.
Reference line 39-40 in response to your comment.
- Introduction: Please mention the wider implications of TBI, in terms of healthcare costs, lost labor, etc
Reference line 28-29 in response to your comment.
- Introduction: Line 33: “Many of these classification tools also tend to be broad in nature” Is this an issue with the classification or the nature of concussion which is not associated with a specific and unique symptomatic presentation? Of course having a precise sign/symptom would be ideal, but how possible is identifying and implementing such specific signs/symptoms in the healthcare setting? I suggest caveats from a clinical perspective should not be mixed with logistics caveats.
The issue the authors are pointing out in this line has to do with the fact that concussion symptom batteries focus on a broad array of symptoms which could be caused by other pathophysiology. This is serving as pretext for the discussion of identifying concussion-specific diagnostic strategies.
- Introduction: Lines 36-28: “The diagnostic imprecision and subjectivity associated with concussion can be especially challenging when considering concussion in athletics, as many athletes may be motivated to subvert concussion detection to remain eligible for participation[7].” This sentence is out of place and it is not clear why athletes are highlighted and focused. It may mislead the reader that the manuscript is about concussions in athletes. If the authors would like to mention athletes, they need to mention other cohorts as well to provide a balanced argument.
Thank you for this thoughtful critique, the authors agree that this line is out of place. Though this paper is not specifically about sport-related concussion, the authors do believe that this issue with subjective concussion diagnostics is a salient one worth mentioning. In response to your comment we have amended the statement in question. Please reference lines 41-44 and line 48 for the changes made.
- Introduction: Lines 41-42: “… also have imperfect reliability and validity, making them inadequate for standalone use in return-to-play decisions” Please provide more specific examples to make such deficiencies clear.
Thank you for this thoughtful suggestion. We reference the paper “Review of Assessment Scales for Diagnosing and Monitoring Sports-related Concussion” by Dessy et. al. as reference 8 at the end of that sentence for this purpose. That citation toughly details the shortcomings with each of the referenced tests and explains why they are specifically inadequate for return to activity decisions. Detailed discussion of the commonly relied-upon symptom batteries is outside the scope of this review.
- Line 41: Please replace “imperfect” with “inadequate” as perfection is rarely achieved in clinical settings, and in the semantic discussion, perfection can be quite subjective!
Reference line 43 in response to your comment.
- Introduction: Please mention briefly the signs and symptoms of concussion.
Reference line 32-35 in response to your comment.
- Line 51: “post-concussion symptoms” Please provide more specific examples
Reference line 32-35 in response to your comment.
- Lines 63-64: “Visual complaints are common in concussion, with a reported prevalence as high as 69% in adult patients” What is the mechanism of injury causing visual complaint in concussion? To achieve a thorough understanding, explaining such mechanisms can help establish diagnostic and prognostic guidelines.
Thank you for this thoughtful suggestion. Please see the addition of highlighted text in lines 69-70 in response to your comment. While detailed discussion of mechanisms behind this phenomenon is outside the scope of this review, we appreciate that being more specific is additive to the paper.
- Line 66: remove “all”
Thank you for this recommendation. The word “all” has been removed.
- Line 76: “. Specific diagnostic criteria for CI are inconsistent within the existing literature” If such inconsistencies exist, how is this going to be incorporated into concussion diagnosis? Please clarify
Please reference the addition of lines 101-103. The authors have emphasized that standardization will be key for application of this measurement in an objective manner in further study.
- Line 180: Nystagmus can also overlap with other conditions. Rightly, the authors criticized that some current criteria for concussion can overlap with other conditions, such as dementia. Similarly, nystagmus can overlap with other conditions. So how is the alternative better? Please clarify.
Please reference line 478-483, which the authors have added in order to emphasize the fact that none of the discussed tools will be sufficient for diagnosis on their own. With regard to the specific discussion of nystagmus, this phenomenon is not specifically useful for mTBI, as the authors mention that its diagnostic value lies in the fact that it can rule out concussion in favor of more severe injury if it presents acutely following a head injury. The authors also specifically mention in lines 192-193 that it only presents diagnostic value if it occurs spontaneously following head trauma, as it would be unlikely to be related to any concomitant disorder.
- Table 1: Adding a column titled “References” to mention papers that provided information can help readers trace the information.
Thank you for your suggestion. Though we agree that it would be additive to the thoroughness of the table, adding an additional column is not feasible due to formatting constraints. The addition of a column would make the other columns so narrow that the information would become more challenging to interpret.
- Lines 210 and 212: using two “however” does not help the flow. Please rewrite this section.
Reference line 218-219 in response to your comment.
- Line 243: “One challenge with DTI …” Another challenge of DTI is accessibility to such technology in low and middle income countries, where TBI and concussion is more prevalent. So to achieve a universal patient outcome, such factors need to be considered.
Thank you for your thoughtful comment, and we agree that the equitable distribution of resources to ensure adequate care is available in low-income countries will be an important consideration with regard to application of these methods. However, we believe that this discussion is outside the scope of this specific review.
- Adding a figure to summarize the neuroimaging in concussion approaches, such as DTI, tb-fMRI, DSC, etc, can improve the quality of the manuscript. Table 2 is fine, but the manuscript needs some figures to visualize the discussed concepts.
Thank you for your thoughtful comment, however we feel the tables utilized in this manuscript adequately present the information discussed and the authors are prioritizing standardization, utilizing the same format for tables throughout the paper.
- Line 392: The usage of AI is revolutionizing the healthcare system and mentioning such an important topic is crucial for this manuscript, so I commend the authors for mentioning this topic.
The authors thank you for your encouraging comment.
- Line 392-393 “Artificial neural networks have shown significant promise in the area of interpreting medical imaging studies and have been a topic of immense interest in the recent medical literature.” This sentence needs a citation and the following paper can be used:
Neurosurgery and artificial intelligence. AIMS Neurosci. 2021 Aug 6;8(4):477-495. doi: 10.3934/Neuroscience.2021025. PMID: 34877400
Thank you for the reference. We have added it as a citation per your comment.
- Biochemical Tissue Concentrations section is significantly shorter compared to the other two sections, whereas there is plenty of evidence to expand on.
The authors agree that the discussions in all three of the main topical sections of this paper are discrete in nature and not completely comprehensive. For the blood-based biomarkers specifically, the intent was to discuss a select number of specific markers with substantial evidentiary support and which will be feasible for clinical application in the shortest timeline. We understand your concern with this approach, and have addressed this limitation in lines 460-469.
- Table 3: Abbreviations should be added to the Table footnote.
We have added a table of all abbreviations used in the paper to the end of the paper, before the reference list, as well as spelled out all acronyms used in Table 3.
- Table 3: Adding two columns mentioning the advantages and disadvantages of each metabolite can be helpful.
The authors have updated the columns in Table 3 to include an “Advantages” column. The addition of a fifth column is unfortunately unfeasible for to formatting constraints, as an additional column would distort the words of the existing columns such that the information would become difficult to interpret.
- Similarly, information in Figure 3 can be summarized into a figure.
Thank you for your thoughtful comment, however we feel that the tables utilized in this manuscript adequately present the information discussed and the authors are prioritizing standardization, utilizing the same format for tables throughout the paper.
- Line 430: “FDA” has been used in line 388, whereas the abbreviation is introduced in line 430. Care should be taken.
Thank you for the observation, it has been corrected.
- Conclusion: Line 453: I would suggest removing the word “extremely” as it is not the case.
Thank you for the observation, it has been removed.
- Line 465: “Consolidating a selection of these tools…” The usage of AI can be highlighted here for making diagnostic and prognostic models.
Thank you for the recommendation, reference line 520 for this correction.
Round 2
Reviewer 1 Report
I appreciate the Limitations and Directions additions.
Reviewer 4 Report
The authors have addressed my comments, and I am happy with this manuscript to proceed further.